# Design and Development of Oleoresins Rich in Carotenoids Coated Microbeads

**Monica Trif** [1,*] , **Dan Cristian Vodnar** [1] , **Laura Mitrea** [1] , **Alexandru Vasile Rusu** [2,*] and **Claudia Terezia Socol** [3]

1   Department of Food Science and Technology, Institute of Life Sciences, University of Agricultural Sciences and Veterinary Medicine, Cluj-Napoca, Calea Mănăştur 3-5, 400372 Cluj-Napoca, Romania; dan.vodnar@usamvcluj.ro (D.C.V.); laura.mitrea@usamvcluj.ro (L.M.)
2   Biozoon Food Innovations GmbH, Nansenstrasse 8, 27572 Bremerhaven, Germany
3   CENCIRA Agrofood Research and Innovation Centre, Ion Meşter 6, 400650 Cluj-Napoca, Romania; clausocol@yahoo.com
*   Correspondence: monica_trif@hotmail.com (M.T.); rusu_alexandru@hotmail.com (A.V.R.); Tel.: +49-172-41-86193 (M.T.)

**Abstract:** The aim of this study was to encapsulate the oleoresins rich in carotenoids extracted from sea buckthorn (*Hippophae rhamnoides*) fruits into a blend of sodium-alginate and κ-carrageenan microbeads (2% $w/v$) coated by a sodium-alginate (2% $w/v$) layer prepared using an ionotropic gelation technique with calcium chloride (2% $w/v$) by dropping method. The fresh obtained coated microbeads had a "fried eggs" like appearance with a size distribution ranging from 4 to 6 mm. The coated microbeads were analyzed for their SEM and fluorescence. The encapsulation efficiency was 92%. Their stability was investigated by evaluation of the physical integrity performance in aqueous media with different pH to mimic the gastrointestinal tract for 24 h at 37 °C under laboratory conditions. The results demonstrated the limitation of the coated microbeads swelling ability under pH 7. The coated microbeads could be a good tool to guarantee oleoresins rich in carotenoids stability and colon delivery. The present study shows an attractive encapsulation system of oleoresins, in order to obtain stable products for further applications.

**Keywords:** alginate; coated; carrageenan; microbeads; crosslinking; carotenoids

## 1. Introduction

Several scientific studies confirm the presence of many active ingredients in the extract of common sea buckthorn (*Hippophaes rhamnoides*) fruits obtained by cold extraction [1–5]. Oleoresins can be extracted from various plant sources and from different plant parts. Oleoresins in food are responsible for the flavor of spices and herbs, but at the same time provide color (used as dyes and pigments), e.g., oleoresin from paprika and turmeric. Compared to essential oils they are lipophilic, meaning that they dissolve in fats, oils and lipids, which is the property that provides manufacturers with different options for food formulation.

There is a growing European market for oleoresins. Increasing competition for the spices, herbs and other raw materials for oleoresin extraction stimulates European importers, and others, to search for new sources. Nowadays the European industry increasingly uses formulas with natural ingredients. Therefore, manufacturers are increasingly searching for ways to produce foods and flavorings from natural ingredients. For this trend, oleoresins are some of the most important natural ingredients. There is a wide variety of oleoresins, which gives them many possibilities to formulate new or improved natural foods and flavorings. The only disadvantage, as for most of the natural

ingredients, is that oleoresins can vary in their quality and may affect the taste of the final product. As a result, food and flavorings manufacturers have very strict specifications for oleoresins [6].

Alginate is a thickener and gelling agent. As a thickener, it provides the required texture and is easily handled. Carrageenans are a family of linear sulphated polysaccharides, three main types of carrageenans are known: kappa (κ)-, lambda (λ)-, and iota (ι)-, depending on the number and the position of the ionic sulphate groups in the structure of the molecule [7].

The presence of a suitable cation, typically potassium, or calcium is an absolute requirement as well for gelation of the alginate and carrageenans, especially kappa to proceed [8,9]. Crosslinking gelation is based on the capability of polyelectrolytes to crosslink in the presence of counter ions [10,11].

Encapsulating the bioactive compounds extracted from plants allows preserving them for use as nutritional supplements or as ingredients for healthy food products [12,13].

Our main objectives were to initially extract the oleoresins rich in carotenoids from sea buckthorn (*Hippophae rhamnoides*) fruits, and to encapsulate them into a sodium-alginate and κ-carrageenan complex microbeads coated by sodium-alginate layer prepared using ionotropic gelation technique with calcium chloride as hardening bath by dropping method. Their physical integrity performance in aqueous media with different pH to mimic the gastrointestinal tract for 24 h at 37 °C under laboratory conditions has been tested.

## 2. Materials and Methods

### 2.1. Chemicals

All solvents and other chemicals used were analytical grade Sigma Aldrich (Cluj-Napoca, Romania). The sodium-alginate and κ-carrageenan were provided by Danisco Ltd. (Cluj-Napoca, Romania).

### 2.2. Extraction of Oleoresins from Sea Buckthorn Fruits (Hippophae rhamnoides)

The fresh sea buckthorn (*Hippophae rhamnoides*) fruits were purchase from a small village from Bistrita County, Romania. The fresh fruits were washed, and broken up with a cuter, followed by centrifugation at 20° and at 6000 rpm through a sieve with holes. The holes diameter was 1.5 mm.

To extract the oleoresins the developed procedure was the following: (1) on a first step, three fractions that are mainly a paste, a clear juice and a dry part were obtained. The dry part consisted of sea buckthorn fruits seeds and skin; (2) on the second step the extraction using organic solvents was performed according to Figure 1.

The first mixture of solvents (solvent mixture 1) prepared consisting of methanol: petroleum ether: ethyl acetate in a ratio 1:1:1, was mixed with the paste fraction in a ratio 2:1. This mixture was stirred with a magnetic stirrer for 30 min, and left for extraction over the night at room temperature covered by an aluminum foil.

On the next day, the solvent with oleoresins extracted were removed, and the remaining part material after the extraction was further subjected to extraction with a second mixture of solvents prepared. Ethyl acetate: petroleum ether (solvent mixture 2) in a ratio 1:1 was used for the second extraction. The oleoresins were removed and put together with the oleoresins fraction from the extraction with the first mixture of solvents.

The oleoresins were washed with saturated NaCl solution and the soaps were eliminated. Then the oleoresin fraction was treated with $Na_2SO_4$ and vacuum filtered to eliminate all the water residues. The final step is the evaporation when Soxhlet apparatus removed all the solvents that remained and therefore the pure oleoresins fraction was obtained. Extractions were carried out near the boiling point of the respective solvents.

All experiments were performed under dim light as much as possible and with equipment covered with aluminum foil to prevent possible carotenoids photo-oxidation.

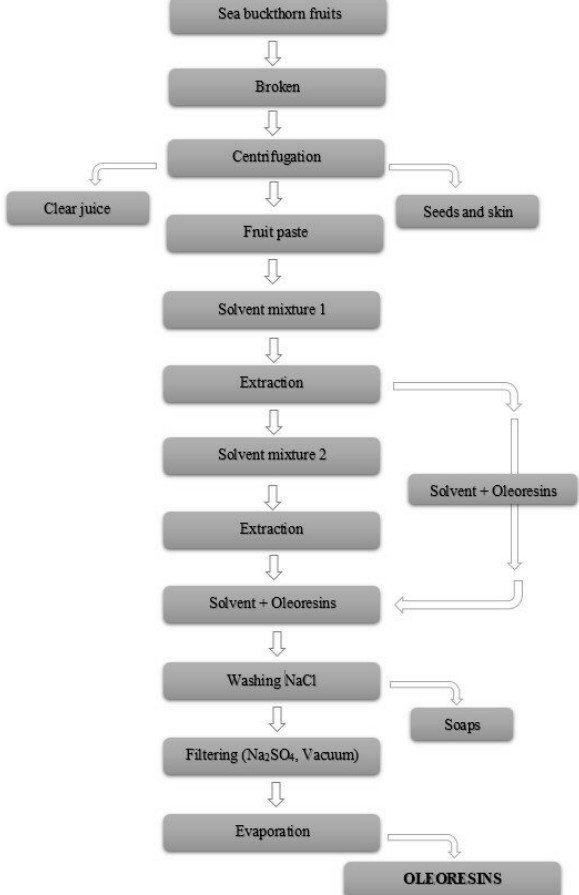

**Figure 1.** Extraction design of oleoresins.

## 2.3. Fractions Analysis

The pH and total acidity of the obtained fractions were determined according to protocols described by Cioroi et al. [14]. Total acidity was measured by direct titration of a 10 mL sample with 0.1 M NaOH standardized solution and the results were expressed as malic acid.

The obtained pure oleoresins fraction was characterized under light microscopy, with a Zeiss high performance microscope (Zeiss, Cluj-Napoca, Romania) for particle size and aspects.

## 2.4. Analysis of Biopolymers

The sodium-alginate and κ-carrageenan rheology was measured on a penetrometer (PCE Instruments, Bremerhaven, Germany). The penetrometer was equipped with a cone assembly of given weight, which is allowed to sink into a hydrogel solution at a certain temperature. The deeper the cone sinks into the material, the softer the material is. The viscosity on a Haake ViscoTester VT 550 viscometer (Thermo Fisher Scientific, Bremerhaven, Germany) at 20 rpm. The pH was measured using an electrode-based pH meter (Mettler-Toledo, Bremerhaven, Germany), simply without any dilution.

## 2.5. Coated Microbeads Preparation

A mixture of sodium-alginate: κ-carrageenan 60:40 $w/w$ (2% $w/v$) was dissolved in de-ionized water under magnetic stirring for ~30 min at 60 °C and allowed to cool down to ~37 °C. The solution prepared was blended with sea buckthorn oleoresins extracted (5% $v/w$) using a high shear homogenizer (Ultra-Turrax, Cluj-Napoca, Romania; 5000 rpm) to form an oil-in-water emulsion. The emulsion was afterwards dropped (dropping method) into a stirred 2% ($w/v$) solution of $CaCl_2$ concentration 1 mM in water (as hardening water bath), using a peristaltic pump with an injector:

0.4 mm × 20 mm, and stirred for ~1 h. The microbeads were separated from the hardening bath, and were then suspended in a sodium-alginate 2% *w/v* solution for coating. The coated microbeads were formed by taking the microbeads into a plastic pipette and dropping them into a hardening bath of CaCl$_2$ concentration 1 mM in water. After few minutes of curing time the coated microbeads were separated from the hardening bath, washed with de-ionized water, and were put on Petri dishes for "protection" and "conservation".

Coated microbeads were made to dry (dehydrate) for 24 h at room temperature.

All experiments were performed under dim light as much as possible and with equipment covered with aluminum foil to prevent possible carotenoids photo-oxidation until microbeads preparation.

### 2.6. Determination of Encapsulation Efficiency of the Oleoresins

The encapsulation efficiency (EE%) was calculated taking into consideration the amount of β-carotene contained by the oleoresins fraction, before and after encapsulation. The amount of β-carotene of the extract was determined spectrophotometrically at 454 nm using tetrahydrofuran (THF). THF was used as solvent to extract β-carotene from the coated microbeads after the microbeads were crushed using a pestle and mortar [15,16].

The following formulae modified accordingly [17] was used:

$$EE\% = C1/(DF \times C2) \times 100 \tag{1}$$

where C1 = β-carotene concentration in coated microbeads containing oleoresins; C2 = β-carotene concentration in oleoresins before encapsulation; DF = dilution factor of ß-carotene according to the added encapsulation material (in our case DF = 1).

### 2.7. UV-Vis Analysis

The absorption spectra were obtained in a PerkinElmer UV-Vis spectrometer (Cluj-Napoca, Romania). All measurements were performed with the substances inside a 2 mm long quartz glass cuvette. All spectra were recorded at room temperature and the results are the average of 3 runs. The normalized absorption spectrum of carotenoids constituents of oleoresins fraction was performed in the range 300–500 nm. Carotenoids are easily identified at wavelengths between 290 and 500 nm. Quantification can be carried out by comparison with internal standards. [18–20].

### 2.8. Analysis of Coated Microbeads for Encapsulation of Oleoresins

#### 2.8.1. Fluorescence Method

For labeling the encapsulated oleoresins, fluorophore Dil-D-3911 was used. The lipophilic carbocyanines are weakly fluorescent in water but are highly fluorescent and quite photostable when incorporated into membranes. They have extremely high extinction coefficients through modest quantum yields, and short excited-state lifetime (~1 nsec) in lipid environments. Once applied to cells, the dyes diffuse laterally within the plasma membrane, resulting in staining of the entire cell. Transfer of these probes between intact membranes is usually negligible. Dil exhibits distinct orange, green, red and infrared fluorescence, respectively, thus facilitating multicolor imaging. Dil-D-3911 can be used with standard fluorescein and rhodamine optical filters, respectively. Dil-D-3911 1,1′-dioctadecyl-3,3,3′,3′-tetramethylindocarbocyanine perchlorate has molecular formula: C$_{59}$H$_{97}$ClN$_2$O$_4$ and molecular weight: 933.88.

A solution of 1 μM Dil-D-3911 in DMSO (dimethylsulphoside) was prepared. 1 mL of this solution was added to 1 mL of oleoresins, followed by incubation for 1 h at 37 °C. The sample was centrifuged at 4000 rpm, the pellet was collected and Tris buffer pH = 8 was added in a ratio 1:1. The absorbance and emission was measured with the lamp with wavelength ranging from 450 to 700 nm.

### 2.8.2. Scanning Electron Microscopy (SEM)

The physical surface and morphology of the microbeads were determined using a scanning electron microscope (Hitachi S-2700, iMOXS micro-X-ray fluorescence spectrometer with BSE detector, Cluj-Napoca, Romania). Microbeads samples were sputtered with gold and scanned at an accelerating voltage of 15 kV.

### 2.8.3. Microbeads Stability in Different Solutions with Different pH

The obtained coated microbeads stability was measured by evaluation of the physical integrity performance in aqueous media with different pH to mimic the gastrointestinal tract conditions for 24 h at 37 °C under laboratory conditions.

## 3. Results and Discussion

### 3.1. Characterization of Fractions

The pH values of the obtained fractions were in the range 2.7~3.1 ± 0.1, acidity was expressed as malic acid in the range 1.82%~1.9% ± 0.05%. The pure oleoresins fraction extracted was microscopically characterized. The oleoresins dimensions were ranging between 10 and 40 ± 0.02 μm (Figure 2.).

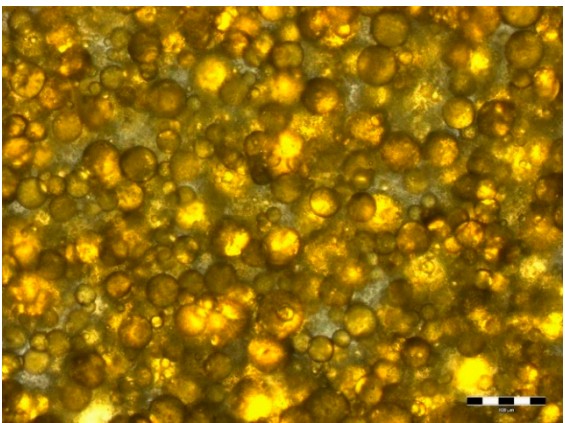

**Figure 2.** Microscopically oleoresins fraction. The scale bar represents 100 μm.

### 3.2. Characterization of Biopolymers

Sodium-alginate break strength in water was 2500–3000 g measured at 20 °C on a penetrometer. The viscosity in a 1% aqueous solution was 200–400 cP measured at 20 °C, on a Haake ViscoTester VT 550 viscometer, at 20 rpm. The pH 6–8.5 measured in a 1% aqueous solution.

κ-Carrageenan strength of a gel at 0.21% in milk: 200–240 g measured after previous solubilization and cooling at 20 °C on a penetrometer. The pH 7–10 was measured in a 1% aqueous solution.

### 3.3. Oleoresins Encapsulation Efficiency Measured by β-Carotene Content

Prior to encapsulation, the β-carotene content was $9.26 \pm 0.05$ mg 100 $g^{-1}$ in the oleoresins fraction, while after encapsulation in the coated microbeads it was $8.52 \pm 0.05$ mg 100 $g^{-1}$, showing that the efficiency of encapsulation was 92%. This result indicates that the obtained coated microbeads has a high encapsulation capacity.

### 3.4. Characterization of Coated Microbeads Containing Oleoresins Encapsulated

The fresh obtained coated microbeads had a "fried eggs" like appearing with a size distribution ranging from 4 to 6 ± 0.02 mm. The size of sodium-alginate-κ-carrageenan complex microbeads containing oleoresins was ranging from 2.5 to 3 ± 0.02 mm, and the coating layer of sodium-alginate

was ranging from 1 to 1.5 ± 0.02 mm. The optimum cross-linker concentration, which resulted in smooth and stable gel microbeads, was applied to microbeads. The yellow-orange color is due to the presence of oleoresins rich in carotenoids (Figure 3).

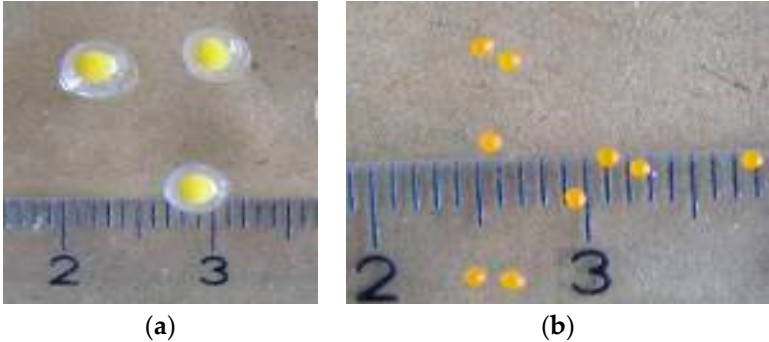

|       |       |
|:-----:|:-----:|
| (**a**) | (**b**) |

**Figure 3.** (**a**) Fresh obtained coated microbeads; (**b**) Coated microbeads dehydrated after 24 h at room temperature.

The coating was produced by forming layer of sodium-alginate around microbeads. Typically, a conventional layer is made followed by additional of a polyelectrolyte of opposite charge, which adsorb to the microbead surface.

In the present study, ionic cross-linking alginate chains prepared the coating layer. We assume that due to sufficient calcium ions being available at the surface of sodium alginate-κ-carrageenan complex microbeads prepared, a further cross-link of the sodium alginate occurred and formed a layer at the surface of the microbeads. The coated microbeads obtained were dropped further into a hardening bath of $CaCl_2$ for ionic cross-linking of sodium-alginate layer, resulting in stable coated microbeads as shown in Figure 3a.

One of the advantages of coated microbeads is that the layer can be engineered to be released from the microbeads under certain environmental conditions. Besides, coated microbeads have been successfully used to increase the oxidative stability of encapsulated bioactives compared to conventional uncoated microbeads.

The diameter shrunk when coated microbeads dried. Similar findings have been reported regarding hydrogel beads [21]. As described in a previous study, the dehydration had a simple power-law time dependence. The loss of mass of a drying coated microbead was governed by the diffusion of saturated water vapor at the surface of the microbeads into the surrounding air at ambient relative humidity [22,23]. The size distribution of the dried coated microbeads ranged from 1 to 1.5 ± 0.02 mm, meaning a volume reduction occurred of up to 70% (Figure 3b).

3.4.1. Fluorescence Characterization

The fluorescence method was used to show the microbeads content prior to coating. The oleoresins were first labeled with the fluorescent Dil-D-3911. The mixture vortexed, encapsulated and then was subjected to microscopy (Figure 4.).

Figure 4a shows microbeads uncoated with oleoresins labeled with Dil-D-3911 trapped in alginate-carrageenan network. As it is known Dil-D-3911 exhibits red-orange fluorescence between 550 and 700 nm. It can observe the orange fluorescent inner tail of the microbeads which is the lipophilic core labeled with Dil-D-3911. The microbeads' wall is colored green because Dil-D-3911 does not label polysaccharides. It is obvious as well from Figure 4a that the microbeads did not contain a homogenous oleoresins quantity.

The Figure 4b shows the UV absorption spectrum and the UV fluorescence emission spectrum of Dil-D-3911 bounded. The maximum absorption is at a wavelength of 545 nm, with the maximum fluorescence emission at a wavelength of 565 nm.

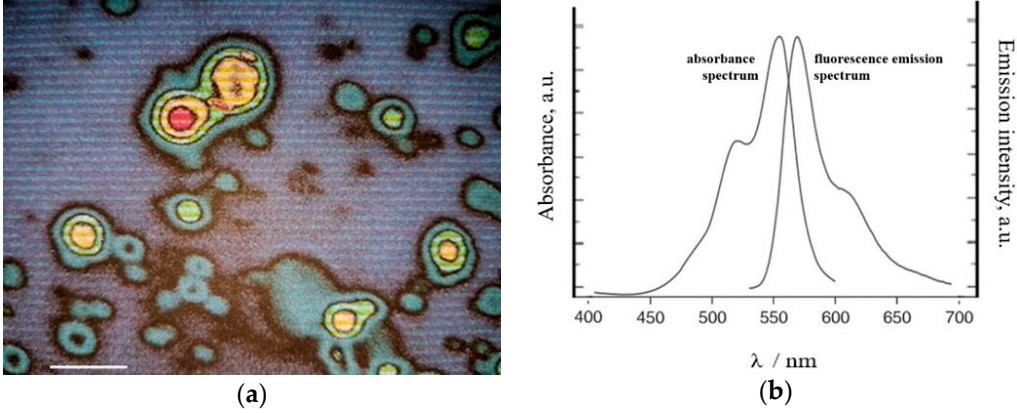

(**a**)  (**b**)

**Figure 4.** (**a**) Microbeads microscopically with fluorescence. The scale bar represents 3 mm; (**b**) UV absorption spectrum and the UV fluorescence emission spectrum of Dil-D-3911 bounded.

### 3.4.2. Microbeads Surface Morphology

The purpose of the scanning electron microscopy study was to obtain a topographical characterization of coated microbeads. Other studies described the hydrogels beads encapsulating natural extracts from plants having quite smooth surface. In this study case, the surface obtained is non regular, due to the oleoresins dispersed over the internal structure (Figure 5a). The SEM pictures of beads revealed that the surfaces were found to be nonporous (Figure 5b).

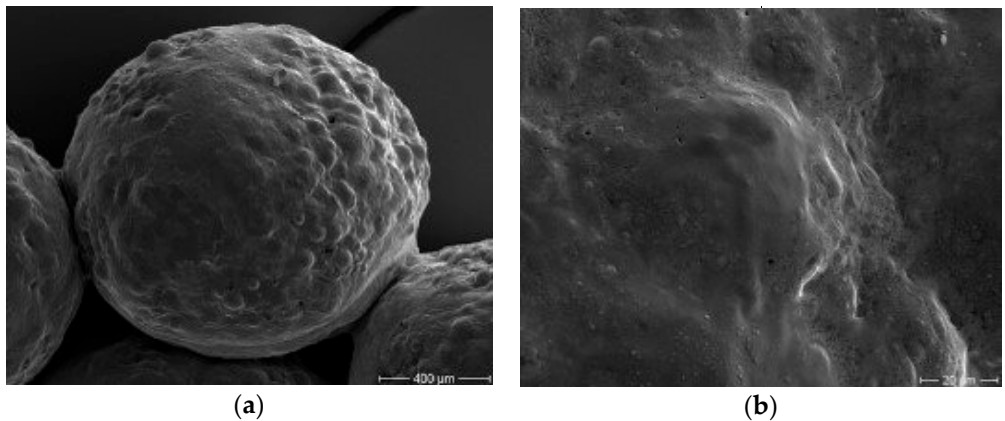

(**a**)  (**b**)

**Figure 5.** (**a**) Scanning electron micrographs of external structure of coated microbeads. Magnification 70×; (**b**) Scanning electron micrographs of surface morphology. Magnification 1000×. The scale bars are shown on the individual photographs.

### 3.5. Microbeads Stability in Different Solutions with Different pH

The coated microbeads stability was measured by evaluation of the physical integrity performance in aqueous media with different pH to mimic the gastrointestinal tract for 24 h at 37 °C under laboratory conditions.

The scheme of using the simulated fluids at different pH was as follows [23,24]:

- pH = 2 to mimic stomach pH: consisted of 0.1 N HCl;
- pH = 4.5 to mimic intestinal fluid: mixing solution pH 1.2 and solution pH 7.4 in a ratio 39:61; pH adjusted to 4.5 ± 0.1.
- pH = 5–6.5 to mimic duodenum and proximal jejunum pH;
- pH = 7.1–7.4 to mimic saliva and colon pH: consisted of 1.074 g $KH_2PO_4$ in 30 mL of 0.2 N NaOH;
- pH = 6.5–8 to mimic large bowel pH.

It was observed that the sizes between different beads varied as shown in Figure 6b. The swelling volume at pH 4 and 6.2 was higher than at pH 2, and a swelling rate from around $2 \pm 0.02$ mm to $4{\sim}5 \pm 0.02$ mm was observed. At pH 7.2 and 7.5 the coated microbeads were completely dissolved and destroyed, therefore formation of two phases can be observed (Figure 6a).

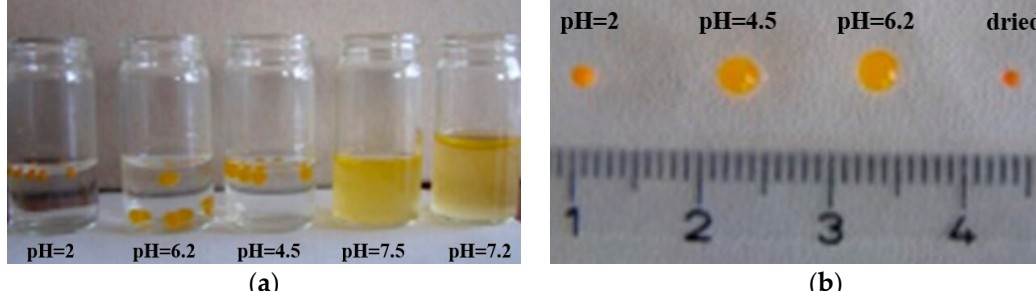

| (a) | (b) |

**Figure 6.** (**a**) Swelling properties in different pH (pH = 2; pH = 6.2, pH = 4.5; pH = 7.5; pH = 7.2, from the left to the right) after 24 h; (**b**) Coated microbeads after 24 h in different pH (pH = 2; pH = 4.5; pH = 6.2, from the left to the right) and dried at room temperature.

The coated microbeads have maintained their physical integrity up to pH 6.2 until reaching a pH higher than 7 and starting to disintegrate. The designed coated system may be suitable for food and nutraceutical applications. The coated microbeads are able to delay the oleoresins rich in carotenoids releasing in an acidic environment and to promote their release in the intestinal part.

Carrageenan, in general, is very stable when gelled at a pH of between 3.5 and 6, at a pH below 3.5, carrageenan is not recommended for use as a gelling agent as it will be unstable. Therefore, to protect a bioactive that is encapsulated, there is a need for developing a complex to help carrageenan maintain its stability below pH 3.5. In the present study a mixture of sodium-alginate and κ-carrageenan has been developed to improve the stability under pH 3.5 of κ-carrageenan. By coating the microbeads obtained using the proposed mixture by a layer of sodium-alginate the stability will be assured, proven by the tests performed.

Up to now, no reports are available which describe the enhanced stability of either sodium-alginate and κ-carrageenan complex-based microbeads, or sodium-alginate and κ-carrageenan complex-based coated microbeads along the gastrointestinal tract. Therefore, our results provide initial evidence of an attractive coating complex.

Researchers have been searching for microencapsulation systems of oleoresin for different applications. Oleoresins successfully extracted from different natural sources were encapsulated in various systems such as: chitosan and alginate for fresh milk preservatives [25]; chitosan-alginate complex, gum arabic and modified starch, gum arabic and soy protein isolate, binary and ternary blends of gum arabic, maltodextrin and a modified starch as wall materials using spray drying [26–29], or as oleoresins of pepper encapsulated in sodium alginate films with essential oils as treatment for infected wounds [30].

Hydrogels have not been used often to prepare gel beads containing oleoresins. Therefore, the present study shows an attractive complex of sodium-alginate-κ-carrageenan coated by a generated sodium-alginate layer, which seems to be a promising encapsulation system of oleoresins rich in carotenoids. Besides, ionotropic gelation has proven to be a very good encapsulation method due to its lower operating costs [31,32].

## 4. Conclusions

The encapsulation of oleoresins extracted from fresh sea buckthorn fruits has been achieved. This study provides initial evidence that successfully extracted oleoresins from sea buckthorn fruits can be encapsulated in sodium alginate-κ-carrageenan cross-linked microbeads coated by a sodium alginate layer. Therefore, ensuring a protection of bioactives from environmental conditions, the results

demonstrate a new strategy for further industrial applications of coated microbeads obtained such as for functional food products, which is an increasingly valued market.

**Author Contributions:** Conceptualization, M.T. and A.V.R.; Methodology, M.T.; Validation, M.T., D.C.V. and C.T.S.; Formal Analysis, M.T.; Investigation, M.T., L.M., D.C.V., A.V.R. and C.T.S.; Data Curation, M.T.; Writing—Original Draft Preparation, M.T. and A.V.R.; Writing—Review and Editing, M.T., D.C.V., A.V.R. and C.T.S.; Visualization, M.T.; Supervision, M.T.; Project Administration, M.T.; Funding Acquisition, M.T., D.C.V., A.V.R. and C.T.S.

**Funding:** This research was funded by the Competitiveness Operational Programme, Priority Axis 1, Action 114, Project Type "Attracting high level personal from abroad" POC-A1-A1.1.4-E_2015 developed with the support of MCI (Acronym: ProGlyCom, Project ID: POC/ID P_37_637, 2016-2020).

**Conflicts of Interest:** The authors declare no conflict of interest.

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
