# Peer review of "Design and Development of Oleoresins Rich in Carotenoids Coated Microbeads"

_coatings, doi:10.3390/coatings9040235_

Round 1
Reviewer 1 Report
1. Line 119 “microbeads were crushed using a mortal”
Using a mortal? Or using a mortar?
2. Line 133 “For labelling the oleoresins encapsulated Dil-D-3911 was used”
What is the Dii-D-3911? What is the labelling mechanism? It is necessary to cite the reference.
3. Line 136 “The absorbance and emission was measured with the lamp with wavelength ranging from 450 to 700 nm.” It is necessary to show the data in the section of result.
4. Line 147 “Results were expressed as a mean value with its standard deviation (mean ± S.D.)”
However, the results were not expressed as mean ± S.D.
5. Line 148 “Statistical analysis was performed with student's t-test and differences
The results did not show the statistical analysis.
6. The morphology of oleosins fraction is sphere in Figure 2. The oleosine fraction is a dry form or in water solution?
7. The quality of Figure 3 and 5 are poor.
8. It is lack of scale bar in Figure 5. Besides, the microbeads showed the red, yellow and green color. What’s the difference between the red and yellow?
Author Response
1. Line 119 “microbeads were crushed using a mortal”
Using a mortal? Or using a mortar?
Response: Correction done: using a pestle and mortar. Thank you.
2. Line 133 “For labelling the oleoresins encapsulated Dil-D-3911 was used”
What is the Dii-D-3911? What is the labelling mechanism? It is necessary to cite the reference.
Response: What is Dil-D-3911 and labelling mechanism has been described. Thank you.
3. Line 136 “The absorbance and emission was measured with the lamp with wavelength ranging from 450 to 700 nm.” It is necessary to show the data in the section of result.
Response: We do not think such data should be relevant as it is basic, as it is not a chemistry or spectroscopically manuscript. Otherwise, we will focus on many fields rather than coating.
4. Line 147 “Results were expressed as a mean value with its standard deviation (mean ± S.D.)”
However, the results were not expressed as mean ± S.D.
Response: Correction done. Thank you.
5. Line 148 “Statistical analysis was performed with student's t-test and differences
The results did not show the statistical analysis.
Response: The section was removed. Thank you.
6. The morphology of oleosins fraction is sphere in Figure 2. The oleosine fraction is a dry form or in water solution?
Response: Lines 80-81: “The final step is the evaporation when all the solvents were removed Soxhlet apparatus and therefore the pure oleoresin was obtained.” – The pure oleoresins fraction is a liquid fraction.
Oleoresins are extracts composed of a resin in solution in an essential and/or fatty oil, obtained by evaporation of the solvent(s) used for their production. (Source: "EXTRACTS", British Pharmacopoeia, 3, 2009, ISBN 978-0-11-322799-0)
7. The quality of Figure 3 and 5 are poor.
Response: The images have been converted to a requested format, which may be subjected to an image quality reduction. We re-converted them hopefully the quality is better.
8. It is lack of scale bar in Figure 5. Besides, the microbeads showed the red, yellow and green color. What’s the difference between the red and yellow?
Response: Description added. Thank you.
Reviewer 2 Report
The proper communication paper deals with the extraction of oleoresins from Hippophaes rhamnoides and their further encapsulation to microbeads prepared by ionotropic gelation using k-carrageenan and sodium-alginate as polymeric matrices.
Unfortunately the proper study deals with many fields and the authors didn’t succeed to present documented results. Furthermore, additional experiments have to be made in order to present a comprehensive study able to be published.
The comments for the proper manuscript are:
1. As reading the title of the manuscript it would be expected that experiments such as dissolution profile of oleoresins from microbeads or cytotoxicity tests or the prepared formulations had been conducted. Hence, although the phrase “in vitro bioavailability” is written in the title, no such experiments appeared and so the word “bioavailability” has to be excluded from the title.
2. In Section 2, a whole paragraph concerning the chemicals used must be inserted (where did the authors percussed them from and which is their purity).
3. In Line 65 the phrase “6000 rot/min” must be replaced by the phrase “6000 round/min”.
4. The paragraph from Line 67 to 70, where the extraction procedure is described, is not at all well written. Even in Figure 1 there is no written the solvent to which oleoresins are extracted. I assume that petroleum ether is the proper one (must be clearly written).
5. In Line 71 the symbol “:” after the word prepared must be deleted.
6. In Lines 71 – 72 the right names of the solvents used are “petroleum ether” instead of “ether petroleum” and “ethyl acetate” instead of “ethylic acetate”. (the same error appears in line 75).
7. The paragraph from Line 78 to 81, where the washing and filtration procedures are described, is not at all well written. Even in Figure 1 there is no written what are the soaps formed, at which temperature did the evaporation took place, the evaporation was conducted by rotary evaporator? (must be clearly written).
8. In paragraph 2.32 where the fraction analysis is described a small paragraph about the pH and the total acidity must be written about what how the researches did this. Only the bibliography reference is not enough.
9. In the same paragraph GC-MS analysis must be used in order to verify that the total amount of organic solvents is removed; ie. Ethyl acetate and petroleum ether. Petroleum ether has b.p. 40-60oC and ethyl acetate has b.p. 70.1oC. It must be clarified that no solvent or approved values from FDA are present in the formulations.
10. In Line 86 the number of the bibliographic reference must be written to the Figure’s title.
11. In line 92 the phrase “9:1 v/v” must be substituted by “90/10 v/v’.
12. In paragraph 2.4 the authors must refer which penetrometer was used, the company, its origin, as well as which penetration device used. In Line 98 the type of viscometer, its origin, and the type of viscosity, i.e. by Brookfield, must be written in the manuscript.
13. In Line 98 the authors wrote “appropriate mobile”. Which is the appropriate mobile? The authors must clarify it.
14. The cahracteristics of k-carrageenan and sodium alginate must be inserted in the manuscript. Both of those must be written in the paragraph reffered in comment 2.
15. In Line 103 the phrase “A mixture of sodium-alginate and k-carrageenan (2% w/v, 6:4 ratio) must be replaced by “A mixture of sodium-alginate : k-carrageenan 60:40 w/w” and the concentration 2%w/v must be inserted to the aqueous solution formed.
16. In Line 104, the temperature where the solution formed must be about 60oC. Otherwise, either modified k-carrageenan was used either there were present agglomerates in the final product.
17. In Line 104, the authors must clarify if mechanical or magnetic stirring was used.
18. In paragraph 2.5 the authors did not mention how did the dehydration process conducted, in an oven or in freeze dryer.
19. In Line 143, the microbeads stability must be performed by friability test and not by physical integrity.
20. In Line 144 the word “transit” must be replaced by the word “track”.
21. Concerning the results and discussion the results are epigraphically described.
22. In line 154, the microscopy can be used to obtain the morphology of the oleoresins. Their average size must be determined by DLS.
23. In paragraph 3.2 the morphology of the peaks are not good; i.e. there are not well separated by other by-products.
24. In paragraph 3.4, the authors found that the efficacy of encapsulation was 92%. How was this succeeded since no surfactant was used during preparation procedure (paragraph 2.5). According to the proper procedure the emulsion formed would not be stable and two phases, one aqueous and one oil phase, would be appeared. This is also verified by Figure 5 and in Line 194, where it refers that there was no homogenized quantity of oleoresins into microbeads. Hence, the value 92% must be measured again and the result must be presented with the standard deviation.
25. As mentioned in comment 18, dehydration process is not referred. A) If oven was used then the temperature of 100oC would probably affect the beads prepared. This could explain the absence of the coating in Figure 4b. If so TGA analysis must be conducted. B) If freeze drying was used the proper temperature must be referred. Freeze drying procedure could also result in the morphology of microbeads prepared, as shown in Figure 6. Since, no homogenized quantity of oleoresins was present into microbeads, net microbeads, by the absence of oleoresins, must also prepared and characterized by SEM in order to be sure that the irregularity in the surface, as observed by SEM, is ought to oleoresins.
26. The authors must explain why after pH value of 7 two phases are formed, i.e. microbeads are destroyed, as shown in Figure 7.a.
27. Finally, swelling ratio must be determined and must be included in the manuscript
Author Response
The proper communication paper deals with the extraction of oleoresins from Hippophaes rhamnoides and their further encapsulation to microbeads prepared by ionotropic gelation using k-carrageenan and sodium-alginate as polymeric matrices.
Unfortunately the proper study deals with many fields and the authors didn’t succeed to present documented results. Furthermore, additional experiments have to be made in order to present a comprehensive study able to be published.
Response: The manuscript has been submitted as Short/Communication/ (preliminary, but significant, results will be considered), rather than a real scientific and comprehensive publication.
We presented the extraction of oleoresins, as we targeted to encapsulate them into a new design system. We believe is necessary to describe it in the present study, otherwise how to mention about, only saying the extracted oleoresins from sea buckthorn fruits. The extraction has not been published to refer to it.
Following the structure of already published communication papers in the Coatings Journal, we presented the preparation of the coated microbeads, the characterization of them and we tested their physical integrity in aqueous media with different pHs. The results demonstrated the limitation of the coated microbeads swelling ability under pH 7, meaning colon delivery guarantee.
Based on the preliminary results obtained and presented in this communication, we will further perform additional tests in order to present later on a comprehensive study on their in vitro stability and bioavailability of encapsulated oleoresins rich in carotenoids.
We strongly believe that the present study shows an attractive encapsulation system of oleoresins. Besides, from our knowledge the oleoresins from sea buckthorn fruits has not been yet encapsulated in any system. Thank you.
The comments for the proper manuscript are:
1. As reading the title of the manuscript it would be expected that experiments such as dissolution profile of oleoresins from microbeads or cytotoxicity tests or the prepared formulations had been conducted. Hence, although the phrase “in vitro bioavailability” is written in the title, no such experiments appeared and so the word “bioavailability” has to be excluded from the title.
Response: The content and title of the communication have been reconsidered. Thank you.
2. In Section 2, a whole paragraph concerning the chemicals used must be inserted (where did the authors percussed them from and which is their purity).
Response: Correction done. Thank you.
3. In Line 65 the phrase “6000 rot/min” must be replaced by the phrase “6000 round/min”.
Response: Correction done. Thank you.
4. The paragraph from Line 67 to 70, where the extraction procedure is described, is not at all well written. Even in Figure 1 there is no written the solvent to which oleoresins are extracted. I assume that petroleum ether is the proper one (must be clearly written).
Response: Correction done. Thank you.
5. In Line 71 the symbol “:” after the word prepared must be deleted.
Response: Correction done. Thank you.
6. In Lines 71 – 72 the right names of the solvents used are “petroleum ether” instead of “ether petroleum” and “ethyl acetate” instead of “ethylic acetate”. (the same error appears in line 75).
Response: Correction done. Thank you.
7. The paragraph from Line 78 to 81, where the washing and filtration procedures are described, is not at all well written. Even in Figure 1 there is no written what are the soaps formed, at which temperature did the evaporation took place, the evaporation was conducted by rotary evaporator? (must be clearly written).
Response: Description of evaporation has been added lines 81-82.
8. In paragraph 2.32 where the fraction analysis is described a small paragraph about the pH and the total acidity must be written about what how the researches did this. Only the bibliography reference is not enough.
Response: Referring to section 2.2 (as section 2.32 does not exists), description has been added lines 86-87.
9. In the same paragraph GC-MS analysis must be used in order to verify that the total amount of organic solvents is removed; ie. Ethyl acetate and petroleum ether. Petroleum ether has b.p. 40-60oC and ethyl acetate has b.p. 70.1oC. It must be clarified that no solvent or approved values from FDA are present in the formulations.
Response: The organic solvents used in the present study were chosen based on a literature review. As they are most commonly used to extract the oils we assume that in the final extract their values are according to the expectations. In any case, we agree that a further analysis has to be performed, as for an industrial application everything has to be FDA approved. Therefore, all necessary analysis for FDA approval will be done at that time. Thank you very much for your recommendation.
10. In Line 86 the number of the bibliographic reference must be written to the Figure’s title.
Response: Only the pH and total acidity of the obtained fractions were determined according to protocols described by Cioroi et al. [14].
The extraction design of oleoresins presented in Figure 1 has been drafted by authors of the manuscript.
11. In line 92 the phrase “9:1 v/v” must be substituted by “90/10 v/v’.
Response: Correction done. Thank you.
12. In paragraph 2.4 the authors must refer which penetrometer was used, the company, its origin, as well as which penetration device used. In Line 98 the type of viscometer, its origin, and the type of viscosity, i.e. by Brookfield, must be written in the manuscript.
Response: Correction done. Thank you.
13. In Line 98 the authors wrote “appropriate mobile”. Which is the appropriate mobile? The authors must clarify it.
Response: Correction done. Thank you.
14. The cahracteristics of k-carrageenan and sodium alginate must be inserted in the manuscript. Both of those must be written in the paragraph reffered in comment 2.
Response: Correction done. Thank you.
15. In Line 103 the phrase “A mixture of sodium-alginate and k-carrageenan (2% w/v, 6:4 ratio) must be replaced by “A mixture of sodium-alginate: k-carrageenan 60:40 w/w” and the concentration 2%w/v must be inserted to the aqueous solution formed.
Response: Correction done. Thank you.
16. In Line 104, the temperature where the solution formed must be about 60oC. Otherwise, either modified k-carrageenan was used either there were present agglomerates in the final product.
Response: Correction done. Thank you.
17. In Line 104, the authors must clarify if mechanical or magnetic stirring was used.
Response: Correction done. Thank you.
18. In paragraph 2.5 the authors did not mention how did the dehydration process conducted, in an oven or in freeze dryer.
Response: As mentioned in the figure where shown, coated microbeads were let to dehydrate for 24 hours at room temperature. We add the details in section 2.5 as well.
19. In Line 143, the microbeads stability must be performed by friability test and not by physical integrity.
Response: Thank you for your suggestion. To our knowledge friability testing is a laboratory technique used by the pharmaceutical industry to test the durability of tablets during transit. In our case it is about coated microgelbeads, we do not consider such test should be applied but in any case we will ask further opinion on.
In this study the aim was to test if the coated microbeads have maintained their physical integrity, being continuously exposed to different pH solutions.
20. In Line 144 the word “transit” must be replaced by the word “track”.
Response: Hope you meant tract… Correction done. Thank you.
21. Concerning the results and discussion the results are epigraphically described.
22. In line 154, the microscopy can be used to obtain the morphology of the oleoresins. Their average size must be determined by DLS.
Response: The oleoresins extracted were microscopically characterized. We could maybe consider later on a primary size distribution obtained from a DLS, but the size distribution obtained microscopically is relevant.
23. In paragraph 3.2 the morphology of the peaks are not good; i.e. there are not well separated by other by-products.
Response: Our target is not the separation of the different carotenoids, therefore, the aimed HPLC analyse was to check the components of the oleoresins.
As not relevant the section was removed. Thank you.
24. In paragraph 3.4, the authors found that the efficacy of encapsulation was 92%. How was this succeeded since no surfactant was used during preparation procedure (paragraph 2.5). According to the proper procedure the emulsion formed would not be stable and two phases, one aqueous and one oil phase, would be appeared. This is also verified by Figure 5 and in Line 194, where it refers that there was no homogenized quantity of oleoresins into microbeads. Hence, the value 92% must be measured again and the result must be presented with the standard deviation.
Response: Indeed, we succeeded to prepare the oil-in-water emulsion (oleoresins fraction-in-hydrogels solution) using no surfactant. The fresh emulsion was afterwards dropped into the hardening bath. The stability on time of the oil-in-water emulsion has not been tested, therefore the “would not be stable” situation has to been tested. Besides, in the referred figure showing the microbeads microscopically fluorescence, the microbeads did not contain a homogenous oleoresins quantity and we can not conclude that means an instability.
25. As mentioned in comment 18, dehydration process is not referred. A) If oven was used then the temperature of 100oC would probably affect the beads prepared. This could explain the absence of the coating in Figure 4b. If so TGA analysis must be conducted. B) If freeze drying was used the proper temperature must be referred. Freeze drying procedure could also result in the morphology of microbeads prepared, as shown in Figure 6. Since, no homogenized quantity of oleoresins was present into microbeads, net microbeads, by the absence of oleoresins, must also prepared and characterized by SEM in order to be sure that the irregularity in the surface, as observed by SEM, is ought to oleoresins.
Response: If we refer again to the figure where shown, coated microbeads were let to dried (dehydrate) for 24 hours at room temperature. We add the details in section 2.5 as well. The oven and freeze drying is out of our scope. Dehydration in a freeze drier (60 °C) or oven (100 °C) would affect the carotenoids stability (oxidation).
The experts’ opinion performed the SEM analysis is that the irregularity in the surface, as observed by SEM, is due to the oil droplets dispersed all over the internal structure, as in general the hydrogel beads have quite smooth surface, as well based on literature review. Oleoresins are extracts composed of a resin in solution in an essential and/or fatty oil.
26. The authors must explain why after pH value of 7 two phases are formed, i.e. microbeads are destroyed, as shown in Figure 7.a.
Response: As described in the text at pH 7,2 and 7,5 the coated microbeads were completely dissolved, therefore the two phases are formed. We added also term destroyed in the text. Thank you.
27. Finally, swelling ratio must be determined and must be included in the manuscript.
Response: From the respective figure where presented it is observable a swelling rate from around 2 to 4-5 mm. Correction done. Thank you.
Reviewer 3 Report
There are many problems from the design of experiments to the analysis of data, and in their presentation. Here are some of those:
1) The paper in general is not really like a scientific publication. It does not provide meaningful conclusions, but merely list the results from a few analyses. This makes the work more like a lab or industrial report, rather than a scientific publication. For instance, what is the final conclusion from the information ending at line 168?
2) Although the equation used in this study is a routine way for reporting the encapsulation efficiency of the system, it should be noted that the main focus of this study, as also insisted in the title, should be the “encapsulation capacity” of the beads. The encapsulation efficiency as measure in the present manuscript does not properly provide this information. EE% as it is may not necessarily improve the bioavailability of the drug.
3) Figure 3 does not provide an acceptable fashion of data presentation for scientific publication at all. The raw data needs to be plotted and reformatted according to the journal’s guidelines.
4) What is the significance of figure 5? Is the caption really meaningful?
5) What significant information is really presented in figure 6 b?
6) Based on the SEM images provided, the beads look to be attached at the surface. Is that correct? If yes, how was this problem solved?
7) Lines 152-155: all single-sentence paragraphs! Why do not you bring them in a paragraph?
8) The information provided in lines 152 and 153 are not easily understandable. What do you mean exactly?
9) The scale bar is not readable at all. The quality of the image itself is also very poor.
10) Regarding figure 7, this may not be the best possible method for the measurement of the swelling ratio of particles. Why routine methods were not used?
11) The manuscript needs considerable grammatical revisions: lines 136, 63, and...
12) Subtitle 2.8.1 has a different font
13) Line 128, should it be absorption or absorbance?
There are many problems from the design of experiments to the analysis of data, and in their presentation. Here are some of those:
1) The paper in general is not really like a scientific publication. It does not provide meaningful conclusions, but merely list the results from a few analyses. This makes the work more like a lab or industrial report, rather than a scientific publication. For instance, what is the final conclusion from the information ending at line 168?
Response: The manuscript has been submitted as Short/Communication/ (preliminary, but significant, results will be considered), rather than a real scientific publication.
The manuscript title and content have been revised. Thank you.
2) Although the equation used in this study is a routine way for reporting the encapsulation efficiency of the system, it should be noted that the main focus of this study, as also insisted in the title, should be the “encapsulation capacity” of the beads. The encapsulation efficiency as measure in the present manuscript does not properly provide this information. EE% as it is may not necessarily improve the bioavailability of the drug.
Response: In the presented manuscript we did not deal with any DRUG, oleoresins are extracted from fruits, the application is food and nutraceutical industry.
To our experience and based on literature review, when doing microencapsulation of natural extracts (different bioactives) from plants in different systems, the encapsulation efficiency (EE%) is calculated. It is calculated as the amount of bioactives of interest encapsulated in microbeads (mb) divided by the bioactives of the solution used for the preparation of microbeads (ms), as shown in equation: EE%=mb/ms x 100. And the results indicate that this type of coated microbeads has high encapsulation capacity.
3) Figure 3 does not provide an acceptable fashion of data presentation for scientific publication at all. The raw data needs to be plotted and reformatted according to the journal’s guidelines.
Response: The section has been removed. Thank you.
4) What is the significance of figure 5? Is the caption really meaningful?
Response: The fluorescence method was used to show the microbeads content. Figure 5 shows microbeads uncoated with oleoresins labeled with Dil-D-3911 trapped in sodium alginate- k-carrageenan network.
5) What significant information is really presented in figure 6 b?
Response: Scanning electron micrographs of surface morphology, as the surface of beads obtained is non regular and found to be non-porous. The experts’ opinion performed the SEM analysis is that should be shown, as in general the hydrogel beads have quite smooth surface, as well as based on literature review.
6) Based on the SEM images provided, the beads look to be attached at the surface. Is that correct? If yes, how was this problem solved?
Response: Microbeads samples were sputtered with gold and scanned at an accelerating voltage of 15 kV, that’s why they are attached.
7) Lines 152-155: all single-sentence paragraphs! Why do not you bring them in a paragraph?
Response: Correction done. Thank you.
8) The information provided in lines 152 and 153 are not easily understandable. What do you mean exactly?
Response: The section 2.3 has been detailed. We believe now the results of the section related to it are understandable. Thank you.
Following reviewers’ recommendation for some other sections as well, if you consider this section as basic analysis and not relevant, we could remove it.
9) The scale bar is not readable at all. The quality of the image itself is also very poor.
Response: The scale bar has been generated, in the description of each figure it has been mentioned the scale bar representation. The images have been converted to a requested format, which may be subjected to an image quality reduction. We re-converted them hopefully the quality is better.
10) Regarding figure 7, this may not be the best possible method for the measurement of the swelling ratio of particles. Why routine methods were not used?
Response: From the respective figure where presented it is observable a swelling rate from around 2 to 4-5 mm. Correction done. Thank you.
11) The manuscript needs considerable grammatical revisions: lines 136, 63, and...
Response: The communication has been revised. Thank you.
12) Subtitle 2.8.1 has a different font
Response: Correction done. Thank you.
13) Line 128, should it be absorption or absorbance?
Response: It is correct: absorption spectrum of carotenoids. Thank you
Round 2
Reviewer 1 Report
1. “ Line 136 “The absorbance and emission was measured with the lamp with wavelength ranging from 450 to 700 nm.” It is necessary to show the data in the section of result.
Author: We do not think such data should be relevant as it is basic, as it is not a chemistry or spectroscopically manuscript. Otherwise, we will focus on many fields rather than coating.”
I can not agree the author’s opinion. How to measure the absorbance and emission? It is not clear.
2. Line 213 “Figure 5 shows microbeads uncoated with oleoresins labeled with Dil-D-3911 trapped”
Figure 5 is image of SEM. How to show the orange fluorescent?
3. Figure 4 is lack of scale bar.
4. The results must be presented with the standard deviation, Ex: efficiency of encapsulation, the β-carotene content
Author Response
1. “ Line 136 “The absorbance and emission was measured with the lamp with wavelength ranging from 450 to 700 nm.” It is necessary to show the data in the section of result.
Author: We do not think such data should be relevant as it is basic, as it is not a chemistry or spectroscopically manuscript. Otherwise, we will focus on many fields rather than coating.”
I can not agree the author’s opinion. How to measure the absorbance and emission? It is not clear.
Response: Description added plus Figure 4(b). Thank you.
2. Line 213 “Figure 5 shows microbeads uncoated with oleoresins labeled with Dil-D-3911 trapped”
Figure 5 is image of SEM. How to show the orange fluorescent?
Response: Line 213 (Now line 221) -> Figure 4. Microbeads microscopically with fluorescence.
Line 213 -> Figure 5 shows microbeads uncoated with oleoresins labeled with Dil-D-3911 trapped in alginate-carrageenan network. As it is known Dil-D-3911 exhibits red-orange fluorescence between 550-700 nm. It can observe the orange fluorescent inner teil of the microbeads which is the lipophilic core labeled with Dil-D-3911. The microbeads’ wall is green colored because Dil-D-3911 does not label polysaccharides. It is obviously as well from Figure 5 that the microbeads did not contain a homogenous oleoresins quantity.
It was a typing mistake Figure 5 instead of Figure 4, we are sorry for that. The entire paragraph is referring to Figure 4, rather than Figure 5. Microbeads uncoated with oleoresins labeled with Dil-D-3911 trapped in alginate-carrageenan network are shown in Figure 4.
Figure 5 is a SEM image of coated microbeads (Line 229->231).
Corrections done. Thank you.
3. Figure 4 is lack of scale bar.
Response: The scale bar added. Thank you.
4. The results must be presented with the standard deviation, Ex: efficiency of encapsulation, the β-carotene content
Response: Corrections done. Thank you.
Reviewer 2 Report
All the changes proposed by the reviewers has been made.
So I propose for the manuscript to be accepted in its present form.
Author Response
We would like to thank you for your effort and time to read and revise our article.